# A History of Targeted Therapy Development and Progress in Novel–Novel Combinations for Chronic Lymphocytic Leukemia (CLL)

**DOI:** 10.3390/cancers15041018

**Published:** 2023-02-06

**Authors:** Matthew Karr, Lindsey Roeker

**Affiliations:** 1Heme-BMT Hospitalist Division, University of California, San Francisco, CA 94720, USA; 2Memorial Sloan Kettering Cancer Center, Division of Hematology/Oncology, New York, NY 10065, USA

**Keywords:** chronic lymphocytic leukemia (CLL), targeted therapy, precision therapy, minimal residual disease, BTK inhibitors, BCL-2 inhibitors

## Abstract

**Simple Summary:**

The treatment landscape for CLL has changed dramatically since the advent of targeted therapies. Studies have shown clear progression-free survival (PFS) benefit of these agents, as well as overall survival (OS) benefit in some instances, when compared with chemoimmunotherapy (CIT). Building on these successes, novel–novel combinations, including doublets and triplets, are under development with study designs exploring fixed and depth-of-response-driven durations. Further studies will be needed to elucidate the relative contributions of agents more clearly in these combinations and the optimal approach when using novel–novel combinations.

**Abstract:**

Over the last 10 years, the traditional treatment paradigms for CLL have been upended as the use of traditional chemoimmunotherapy regimens has declined in favor of novel targeted therapies. Targeted therapies have become the new standard of care in CLL given their superior progression-free survival (and overall survival, in some cases) when compared with chemoimmunotherapy, as well as their improved toxicity profiles. Targeted agents are FDA approved for the treatment of CLL including ibrutinib, acalabrutinib, zanubrutinib, and venetoclax. Importantly, as opposed to traditional chemotherapy regimens, the benefits of these targeted therapies appear to be consistent regardless of high-risk mutational status. In this review, we discuss the pivotal CLL studies of the last decade and the data supporting doublet and triplet novel–novel combinations. We explore the use of new surrogate end points for PFS/OS in targeted therapies such as undetectable minimal residual disease (uMRD) and their potential role in minimizing toxicity by permitting earlier treatment discontinuation. We also highlight areas that warrant further exploration and future studies that may help address some of these key questions.

## 1. Introduction

The treatment of chronic lymphocytic leukemia (CLL) has undergone a paradigm shift as the mainstay has transitioned from CIT to targeted novel agents, including Bruton Tyrosine Kinase inhibitors (BTKi) and the B-cell lymphoma 2 (BCL2) inhibitor venetoclax. These targeted agents are currently approved as monotherapies or paired with anti-CD20 monoclonal antibodies, although their synergistic mechanisms of action have naturally led to the exploration of novel–novel combination therapies [1,2,3]. Novel–novel combinations, predominantly BTKi + BCL2i doublets, have proven to be highly effective with acceptable tolerability. At least for a subset of patients with CLL, these regimens may represent the future of targeted agent-based therapy in CLL.

Herein, we provide data supporting the use of novel agents in CLL. In Section 2 and Section 3, we review the data for single novel agent regimens in CLL, including covalent BTKis and venetoclax, with a focus on the frontline setting. These data have led to the approval of ibrutinib, acalabrutinib ± obinutuzumab, zanubrutinib, and venetoclax ± obinutuzumab. In Section 4, we explore the development of doublet and triplet novel–novel combination therapies, which have been explored as time-limited options with both fixed and depth-of-response-guided duration approaches. Last, in Section 5 and Section 6, we discuss the limitations of the current data and outline ongoing studies that may answer critical questions as we determine how novel–novel combinations of targeted agents fit into the future treatment landscape of CLL.

## 2. Covalent Bruton Tyrosine Kinase Inhibitors

Prior to 2014, the treatment choice for patients requiring their first CLL-directed therapy depended on the age/fitness of the patient. Young (age ≤ 65 years) and fit patients were treated with fludarabine, cyclophosphamide, and rituximab (FCR). Patients aged 65–75 generally received bendamustine and rituximab (BR). Older or unfit patients were often treated with chlorambucil ± anti-CD20 monoclonal antibodies [4]. These CIT regimens were associated with toxicity and limited efficacy, especially in patients with high-risk cytogenetic or molecular features.

Ibrutinib, the first-generation BTKi, was initially approved in 2014 for the treatment of relapsed/refractory (R/R) CLL and for patients with CLL harboring the deletion of chromosome 17p (del17p) based on data from a phase 1b/2 study with efficacy later confirmed in the phase 3 RESONATE study [5,6]. Given its promise in these populations, RESONATE-2 was conducted, a phase 3 trial in which 269 previously untreated CLL patients aged ≥65 and without deletion of chromosome 17p (del17p) were randomized to receive ibrutinib (n = 136) or chlorambucil (n = 133). Ibrutinib was associated with both a progression-free survival (PFS) and overall survival (OS) benefit, data on which the FDA approval for ibrutinib in the frontline setting was granted in 2016 [7]. With an extended 8-year follow-up, ibrutinib continues to demonstrate a PFS benefit with 7-year PFS estimated at 59% for ibrutinib vs. 9% for chlorambucil (*p* < 0.001) and 7-year OS of 78% in the ibrutinib arm [8]. In this study, PFS was not different among patients with and without high-risk genomic features including del11q and unmutated IGHV. This contrasts with CIT, which was traditionally associated with poorer outcomes in patients with high-risk features [9]. With extended follow-up, 58% of patients have discontinued therapy, with toxicity being the most common reason for discontinuation [8]. While highly effective, these data highlight that the toxicity profile of ibrutinib can limit the agent’s use.

Even at the time of this study, chlorambucil was an infrequently used standard of care regimen, prompting additional studies to compare ibrutinib-based therapy with more “relevant” control arms [10]. A041202 was a phase 3 study conducted in treatment-naïve patients ≥ 65 years of age who were randomized to receive ibrutinib monotherapy, ibrutinib–rituximab, or BR. Both arms containing ibrutinib demonstrated significantly superior PFS at two years over BR (87% vs. 88% vs. 74%, *p* < 0.001), although there was no significant difference in PFS between the ibrutinib and ibrutinib/rituximab arms. Thus, this study suggested that the addition of rituximab did not confer an additional benefit over ibrutinib alone. Although the follow-up period was only 38 months, there was no difference in OS observed in any of the three groups [11]. Toxicity observed significantly more frequently in the ibrutinib-containing arms compared with BR included grade ≥ 3 hematologic adverse events (41% with ibrutinib, 39% with ibrutinib/rituximab), grade ≥ 3 atrial fibrillation (17% with ibrutinib, 14% with ibrutinib/rituximab), and grade ≥ 3 hypertension (29% with ibrutinib, 34% with ibrutinib/rituximab) [11].

Ibrutinib was compared with FCR in the phase 3 study E1912, which included patients ≤ 70 years of age without del17p. E1912 randomized 529 patients in a 2:1 ratio to ibrutinib–rituximab (n = 354) and FCR (n = 175) [12]. The ibrutinib–rituximab arm demonstrated a significantly superior PFS compared with CIT (HR, 0.44; *p* < 0.001) as well as an OS benefit (5 year OS 95% vs. 89%; *p* = 0.018). The benefit was observed regardless of IGHV mutational status [13]. Interestingly, while FCR has historically been thought to lead to long-term remissions in a subset of IGHV mutated patients, the PFS curve for the FCR arm in this study has yet to reach a plateau with a median follow-up of 6 years. With a median follow-up of 6 years, the median time on treatment for those on the ibrutinib/rituximab arm was 59 months. The most common reason for discontinuation was toxicity (22% of ibrutinib/rituximab-treated patients discontinued for toxicity). For those who discontinued for reasons other than progression or death, the median PFS following ibrutinib discontinuation was 25 months, suggesting durable benefit for those who had achieved disease control with ibrutinib/rituximab [13].

The UK Flair study was later presented in 2021 and utilized the same study arms as E1912. While this study replicated the PFS benefit with 52.7 months follow-up (4-year PFS 87% for IR vs. 77% for FCR, *p* < 0.001), an OS benefit was not observed. The authors attributed the lack of OS benefit to significantly improved OS in the FCR arm compared with historical norms (94.5% compared with 84.2% for FCR between 2009 and 2012), likely due to improved treatment in the R/R setting [14,15,16].

ILLUMINATE was a large phase 3 study that compared the combination of ibrutinib and obinutuzumab to chlorambucil/obinutuzumab (CO) in 229 patients who were either >65 or ≤65 with significant comorbidities. With 31.3 months of follow-up, this study also demonstrated a clear PFS benefit for ibrutinib/obinutuzumab over CO (median not reached vs. 19.0 months, *p* < 0.001), regardless of risk features (i.e., with del17p, *TP53* mutation, del11q, or unmutated IGHV) [17]. This benefit was maintained with longer follow-up, as median PFS with 45 months of follow-up was not reached for those treated with ibrutinib/obinutuzumab vs. 22.0 months for those treated with CO (*p* < 0.001) [18]. Like the A041202 study, there was no difference in OS between the two groups (HR = 1.08; 95% CI: 0.60–1.97; *p* = 0.793), although crossover design and short follow-up may explain this. Unlike A041202, however, this study did not include an ibrutinib monotherapy arm. As such, the relative contribution of obinutuzumab to this regimen is difficult to ascertain [19]. The most common adverse events observed with the combination of ibrutinib/obinutuzumab were neutropenia (44%; grade ≥ 3 36%), thrombocytopenia (35%; grade ≥ 3 19%), diarrhea (35%), cough (29%), infusion-related reaction (25%), and arthralgia (24%). Grade ≥ 3 atrial fibrillation was observed in 6% of those who received ibrutinib/obinutuzumab with a median treatment exposure of 42 months [18].

Second-generation BTK inhibitors, including acalabrutinib and zanubrutinib, were designed to target BTK more selectively with the aim of minimizing off-target toxicities compared with ibrutinib. Particularly as real-world studies and long-term follow-up from RESONATE-2 demonstrated that drug discontinuation is frequent and intolerance is the most cited reason for discontinuation of ibrutinib, improved tolerability within the class is particularly relevant [8,20]. While frontline studies have demonstrated their efficacy as compared with chemoimmunotherapy regimens, studies in the relapsed/refractory compare these agents with ibrutinib in order to highlight differences in adverse event profiles [21,22].

Acalabrutinib was approved in the frontline setting based on data from ELEVATE-TN, a phase 3 study in which 535 previously untreated CLL patients were randomized to acalabrutinib monotherapy (n = 179), acalabrutinib/obinutuzumab (n = 179), or chlorambucil/obinutuzumab. Both acalabrutinib-containing arms were superior to CO in terms of estimated PFS at 30 months follow-up, with acalabrutinib/obinutuzumab (AO), acalabrutinib monotherapy, and CO at 90%, 82%, and 34%, respectively (*p* < 0.0001). While not powered to detect a difference between acalabrutinib-containing arms, there was a significant difference in PFS between the acalabrutinib monotherapy and acalabrutinib/obinutuzumab arms. A trend toward improved survival for acalabrutinib-containing arms over chlorambucil/obinutuzumab was noted, with 30-month OS rate of 95% for AO, 94% for acalabrutinib, and 90% for CO, respectively, although this was not significant. With 5-year follow-up from the study, the PFS benefit for the acalabrutinib/obinutuzumab arm over the acalabrutinib arm is even more pronounced, and both arms are significantly superior to chlorambucil/obinutuzumab (84% vs. 72% vs. 21%, respectively, *p* < 0.0001). The updated 5-year survival additionally showed a significant OS benefit for the acalabrutinib/obinutuzumab group over chlorambucil/obinutuzumab (estimated 60-month OS of 90% vs. 82%, HR: 0.55; *p* = 0.0474) [23]. With a median of 47 months of follow-up, 25% of acalabrutinib/obinutuzumab-treated and 31% of acalabrutinib-treated patients discontinued therapy with adverse events driving the majority of discontinuations (13% for acalabrutinib/obinutuzumab, 12% for acalabrutinib) [24].

Zanubrutinib was FDA approved in January 2023 for the treatment of CLL. The SEQUOIA study, which randomized frontline patients without del17p to receive zanubrutinib or BR, demonstrated that PFS was significantly superior for zanubrutinib vs. BR (24-month estimate 86% vs. 70%, *p* < 0.0001). No difference in OS was observed. The study additionally included a cohort of patients with del17p, all of whom were treated with zanubrutinib. With a median of 31 months of follow-up, the estimated 24-month PFS was 89%, and OS was 94%. While follow-up is still relatively short, atrial fibrillation was observed in only 3% and grade ≥ 3 bleeding in 5–7% of those treated with zanubrutinib. Eight percent of those randomized to receive zanubrutinib discontinued for adverse events. The most common adverse event necessitating zanubrutinib discontinuation was COVID-19 (2%) [25].

Second-generation BTK inhibitors have been directly compared with ibrutinib, allowing for clarity regarding differences in toxicity profile. ELEVATE-RR was a noninferiority trial of acalabrutinib vs. ibrutinib in R/R CLL patients with either del17p or del11q. With a median follow-up of 40.9 months, acalabrutinib was associated with lower rates of atrial fibrillation (9.4% vs. 16.0% with ibrutinib; *p* = 0.02), hypertension (9.4% vs. 23.2% with ibrutinib), and treatment discontinuation due to adverse events (14.7% vs. 21.3% with ibrutinib). This study further demonstrated that median time to onset of atrial fibrillation was longer with acalabrutinib (28.8 months vs. 16 months with ibrutinib), which may be significant for future fixed duration strategies [21].

The ALPINE trial was an open-label, noninferiority study that compared zanubrutinib 1:1 with ibrutinib in patients with R/R CLL, regardless of cytogenetic status. In a preplanned interim analysis, the study met its primary end point as zanubrutinib was associated with a superior overall response rate (ORR) compared with ibrutinib (78% vs. 63%, *p* = 0.0006) [22]. With a median follow-up of 30 months, zanubrutinib is additionally associated with a PFS benefit (24-month PFS 80% vs. 67%, *p* = 0.0024) [26]. Treatment discontinuation due to adverse events was less common in patients treated with zanubrutinib (15% vs. 22% with ibrutinib). Cardiac adverse events were less common in patients treated with zanubrutinib (21% vs. 30% with ibrutinib), and zanubrutinib was associated with a lower rate of atrial fibrillation (5.2% vs. 13.3% with ibrutinib; *p* = 0.0004). Similar to the SEQUOIA study, a higher incidence of COVID-19-related pneumonia was seen in the zanubrutinib arm (7.1%) than the ibrutinib arm (4.0%) [26].

## 3. Venetoclax

The BCL2 inhibitor venetoclax was initially approved for patients with relapsed/refractory CLL who had del17p based on results of a single-arm phase 2 study [27]. Later, the combination of venetoclax and rituximab was approved for relapsed/refractory CLL regardless of del17p status based on data from MURANO. In this phase 3 study, a 2-year fixed duration combination of venetoclax/rituximab was compared with 6 months of BR in patients with relapsed/refractory CLL. This study demonstrated a significant PFS (median 53.6 months vs. 17.0 months, *p* < 0.0001) and OS (5-year estimate 82% vs. 62%, *p* < 0.0001) benefit for venetoclax/rituximab over BR [28].

Venetoclax was evaluated in the frontline setting through the large phase 3 study CLL14, an open-label study that randomized 432 treatment-naïve patients in a 1:1 ratio to venetoclax/obinutuzumab versus chlorambucil/obinutuzumab. Venetoclax/obinutuzumab demonstrated significantly superior PFS over chlorambucil/obinutuzumab at 24 months (88.2% vs. 64.1%, *p* < 0.001) without any significant OS difference between the two groups [29]. These data allowed for regulatory approval for VO for previously untreated CLL in May 2019. The most recent 5-year follow-up data from the CLL14 study continue to demonstrate significantly superior PFS rates of the VO arm (62.6% vs. 27%, *p* < 0.0001). Additionally, there was a trend toward improved survival at 5 years (81.9% VO vs. 77% CO), although this trend was not significant (*p* = 0.12) [30]. With this combination, 53% (112/167) experienced grade 3/4 neutropenia, while 18% (37/167) had grade 3/4 infection, and 5% (11/167) had febrile neutropenia. Five deaths (2.4%) were observed during treatment; four of these five deaths were due to infections [29]. Table 1 below summarizes key information from pivotal frontline single novel CLL studies, including those involving BTKis as well as venetoclax.

## 4. Novel–Novel Doublet and Triplet Combinations

Given their complementary mechanisms of action, it was hypothesized that BTK and BCL-2 inhibition would act synergistically. Specifically, BTKis mobilize CLL cells from their microenvironment into circulation, while venetoclax induces apoptosis via BCL2 inhibition most effectively in the peripheral blood.

The first proof of concept combining the two for previously untreated CLL was a phase 2 single-arm study by Jain et al. [31]. Eighty patients who were either ≥65 years of age or had high-risk features were treated with three cycles of ibrutinib followed by 24 cycles of combined ibrutinib/venetoclax (I + V) [31]. This study demonstrated that the administration of ibrutinib prior to the addition of venetoclax successfully reduced the risk of tumor lysis syndrome (TLS), reducing the risk category from high risk in 80% of patients and medium risk in 48% of patients. This study also demonstrated the ability of combination therapy to induce undetectable minimal residual disease (uMRD). In earlier phase 3 monotherapy studies, PFS was largely used as the primary end point. However, this phase 2 study emphasized uMRD as an end point based on the rationale that uMRD status accurately predicted improved PFS and OS in CIT-treated patients [32]. Achieving uMRD was rare in studies of BTKi monotherapy with uMRD rates in earlier studies ranging from 1% in ibrutinib-treated patients in the A041202 study to 7% for acalabrutinib-treated patients in ELEVATE-TN [11,23]. With median follow-up of 33.8 months, uMRD in bone marrow (BM) of I + V was 56% (45/80) after 12 cycles and 66% (53/80) after 24 cycles. The most common grade ≥ 3 adverse events included neutropenia (48%), atrial fibrillation (10%), and hypertension (10%). The most common lower-grade adverse events included easy bruising (60%), arthralgia (48%), and diarrhea (41%). Four patients experienced neutropenic fever, and hospitalization was required for infectious complications in an additional nine patients. Dose reductions were required for ibrutinib in 44% (35/80) and venetoclax in 24% (18/75) [33].

Updated data (median follow-up of 48.8 months) included the original 80 patients and a 40-patient expansion cohort and demonstrated similar results with uMRD in BM in 52% (62/120) after 12 cycles and in 64% (77/120) after 24 cycles. The 4-year PFS rate was 94.5%, and the OS rate was 96.5%. Of the 77 patients with uMRD after 24 cycles, 94.8% (73/77) discontinued all therapy. With a median of 23.9 months post-cycle and 24 off therapy, only 14.2% (11/77) had recurrent blood minimal residual disease (MRD), with only one requiring CLL therapy [34]. Thus, uMRD may be a meaningful surrogate end point for PFS/OS, possibly helping predict which patients may forego the toxicity associated with indefinite treatment. Of the 24 patients who did not achieve uMRD at the completion of 24 cycles, 23 continued ibrutinib, and 18/23 received a third year of I + V. Of these 18 patients, 61% achieved uMRD with a third year of combination therapy [34].

I + V was further studied in the frontline setting through the CAPTIVATE, GLOW, and UK Flair trials. CAPTIVATE was a phase 2 clinical trial that enrolled a fixed duration (FD) cohort (n = 159) and an MRD cohort (n = 164). Patients ≤ 70 years of age with previously untreated CLL in both cohorts received three cycles of ibrutinib lead-in followed by 12 cycles of I + V. Patients in the FD cohort were observed after the completion of 15 cycles. In the MRD cohort, randomization at the end of 15 cycles was based on MRD status. Patients with uMRD were blinded and randomized to ibrutinib maintenance or placebo. In patients with detectable MRD, patients were randomized to open-label ibrutinib monotherapy or continued I + V.

The primary end point for the FD cohort was complete response (CR) rate, specifically in patients without del17p (n = 136, 88%). At 27.9 months follow-up, a CR rate of 56% with I + V was observed, a rate considered promising given previously observed CR rates associated with CIT [35,36]. The CR rate was consistent in the all-treated population (55%) and in patients with del(17p) or TP53 mutation (56%). Secondary end points of significance included 24-month estimated PFS (95% in all-treated, 96% in patients without del(17p), and 84% in patients with del (17p)) and 24-month OS rate (98% in all-treated, 98% in patients without del(17p), and 96% in patients with del (17p)). uMRD was achieved in peripheral blood (PB) for 77% and in BM for 60% of all-treated patients, with similar rates observed in patients with and without del(17p). The most common grade ≥ 3 adverse events included neutropenia (33%), infections (8%), and hypertension (6%) [35].

The primary end point for the MRD cohort was disease-free survival (DFS), an end point that included MRD relapse in addition to disease progression and death. After 12 cycles of I + V, 86 patients (58%) had uMRD in both BM and PB and were randomized 1:1 to ibrutinib (n = 43) or placebo (n = 43). After 31.3 months of follow-up, DFS was 100% in the ibrutinib group and 95% in the placebo group (*p* = 0.15). uMRD was maintained for those treated with placebo (84% in BM and 81% in PB) and ibrutinib (77% in both PB and BM). The high DFS rate and maintenance of uMRD in the placebo group suggests that uMRD may be a useful end point for I + V time-limited therapy. The rates of hypertension were higher for those who continued ibrutinib post-randomization (21% vs. 9% in the first year post-randomization, 24% vs. 12% in the second year, and 27% vs. 16% in the third year). Grade ≥ 3 infections were observed in 5% of both groups prior to randomization, while grade ≥ 3 infections during the first, second, and third years after randomization were experienced by 7% vs. 2%, 10% vs. 2%, and 7% vs. 5% of those randomized to ibrutinib vs. placebo, respectively [37].

Among those with persistent MRD randomized to receive continued ibrutinib (n = 31) or I + V (n = 32), the 36-month PFS was identical at 97%. The uMRD rates deepened after randomization in both arms, although greater improvements in uMRD rates were observed in those who continued I + V. At 2 years post-randomization, 48% achieved uMRD in PB and 42% in BM when receiving continued ibrutinib, while uMRD was achieved in PB for 69% and BM for 66% of those treated with I + V [38]. Notably, the patients included in CAPTIVATE were young and fit (aged ≤70 years with creatinine clearance (CrCL) ≥ 60 mL/min), so it is unclear how well these results will generalize to an older or frail population [35].

The phase 3 GLOW study evaluated the combination of I + V in an older patient population with more comorbidities, specifically patients ≥ 65 years of age or 18–64 years of age with a CIRS score greater than 6 or CrCl of ≤70 mL/min. Patients (n = 211) were randomized to I + V (n = 106) or CO (n = 105). I + V treatment consisted of a three-cycle lead-in of ibrutinib followed by 12 cycles of combination therapy, and CO patients received six cycles of chemoimmunotherapy. The primary end point of the study was PFS. The estimated PFS at 30 months was 80.5% in the I + V arm and 35.8% in the CO arm (*p* < 0.0001). In this older and more comorbid population, grade ≥ 3 adverse events occurred in 76% (80/106) of the I + V treated patients, most commonly grade 3/4 neutropenia (35%), infections (15%; an additional two patients [2%] experienced fatal pneumonia), diarrhea (10%), hypertension (7.5%), and atrial fibrillation (7.5%) [39].

A key secondary end point of uMRD (10^−4^ sensitivity) also significantly favored I + V in both BM (55.7% vs. 21%) and PB (84.5% vs. 29.3%) at the time of primary analysis (median of 27.7 months follow-up). Three months after the end of treatment (EOT + 3), I + V achieved higher uMRD rates compared with CO in BM (51.9% vs. 17.1%, *p* < 0.0001) and in PB (54.7% vs. 39.0%, *p* = 0.0259), with good concordance (92.9%) in PB/BM uMRD in the I + V arm. The authors also analyzed uMRD at a more stringent threshold of 10^−5^ sensitivity to attempt to clarify the depth of uMRD that could be achieved in both arms. With this greater sensitivity, patients treated with I + V achieved higher rates of uMRD compared with CO (40.6% vs. 7.6%), with PB/BM uMRD concordance of 90.9% (40/44). Of the patients with uMRD at 10^−4^ sensitivity, 79.3% of them achieved uMRD 10^−5^ in PB and 78.2% in the BM. uMRD with 10^−5^ sensitivity was maintained from EOT + 3 to EOT + 12 for 80.4% (37/46) of those treated with I + V compared with 26.3% (5/19) for those treated with CO [40]. The most recent analysis with 34.1 months of follow-up showed that 77.6% (45/58) of those with uMRD in the I + V arm maintained PB uMRD from EOT + 3 to EOT + 18 vs. only 12.2% (5/41) in the CO arm. The analysis of uMRD kinetics for IGHV unmutated (uIGHV) vs. IGHV mutated (mIGHV) patients demonstrated deeper and quicker responses for patients with uMRD treated with I + V. Specifically, uMRD was achieved after six cycles for 52.2% of uIGHV and 31.3% of mIGHV and at EOT + 3 in 59.7% vs. 40.6% [41].

The UK Flair trial also included I + V (n = 136) and ibrutinib monotherapy (n = 138) arms with the duration of treatment defined as twice the time to uMRD and ranging from 2 to 6 years. An interim analysis at 2 years post-randomization demonstrated that uMRD (10^−4^ sensitivity) was achieved in BM for 51/64 (79.7%) of uIGHV patients vs. 31/55 (56.4%) of mIGHV patients treated with I + V. The uMRD rates were also higher for uIGHV vs. mIGHV patients treated with I + V at 9 months post-randomization (53.1% vs. 34.5% in BM). These data support the notion that uIGHV patients are more likely to achieve uMRD and more likely to do so at earlier time points with I + V therapy than their mIGHV counterparts [42].

These trials demonstrate that regardless of patient age, high-risk mutational status, or comorbidities, I + V combinations induce deep, durable responses for many patients. The data also demonstrate that time-limited therapy is possible with this regimen, although the optimal duration of therapy and the most appropriate end point for therapy are not yet clear. Based on data from CAPTIVATE and GLOW, the European Commission approved the fixed duration combination of I + V in August 2022.

For patients receiving continuous BTKi, the utility of an “add-on” approach to deepen the response and possibly allow for treatment-free observation was tested in a phase 2 study in 45 patients. Patients who had been on ibrutinib for at least 1 year (either in a frontline or an R/R setting), had high-risk features (del(17p), complex karyotype, del(11q), elevated β_2_-microglobulin, or *TP53* mutation), and met criteria for detectable disease without progression added venetoclax to ibrutinib for up to 2 years. The primary end point was uMRD after 12 months of combination therapy. If a patient achieved uMRD on consecutive evaluations, venetoclax was stopped. Ibrutinib could be continued at the provider’s discretion. In the most recent update, 17 of 42 (40%) achieved uMRD at 6 months, and 21 of 33 (64%) achieved uMRD at 12 months [43], rates considered promising given the high-risk features and inclusion of patients with R/R disease [43].

In addition to these doublet studies, triplet therapies including second-generation BTKis, venetoclax, and obinutuzumab are under study. A phase 2 single-arm, open-label study examined the combination of acalabrutinib/venetoclax/obinutuzumab (AVO) in a cohort of 37 all-comer CLL patients with the later addition of a second cohort of 31 TP53 aberrant patients. Acalabrutinib monotherapy was given starting in cycle 1, obinutuzumab was given in cycles 2 to 7, and venetoclax was introduced at cycle 4. Patients continued acalabrutinib/venetoclax combination therapy through at least cycle 15 with the duration guided by the BM MRD status. The primary end point of the study was CR with uMRD in BM (10^−4^ sensitivity) at completion of cycle 15. If patients achieved uMRD at that point, they could discontinue treatment; if MRD positive, they received 24 cycles with the option to discontinue therapy at C25 if BM uMRD. At 27.6 months of follow-up for the all-comer cohort, 38% (14/37) achieved CR with uMRD in the BM, a rate considered unpromising based on the study design. Despite not reaching the primary end point, the best uMRD rates were 92% (34/37) in the PB and 86% in the BM (32/37) with 94% concordance at cycle 16 and 92% at cycle 25. The rates of uMRD and CR were similar regardless of the high-risk mutation status (TP53, IGHV). Eighty six percent (32/37) achieved uMRD in BM and discontinued therapy with 12 (32%) stopping at cycle 16 and an additional 19 (51%) stopping at cycle 25. With a median of 7.6 months off therapy for the all-comer cohort, no progression events were observed [44]. The AVO regimen was associated with hematologic toxicity including grade ≥ 3 neutropenia (43%) and thrombocytopenia (27%). However, infections were rare, with only one patient experiencing grade 3 infection (2.3%). The study later added a cohort of patients with a TP53 mutation or del(17p) to examine the combination’s effect exclusively in high-risk patients [44]. With a median follow-up of 35 months, CR with uMRD was achieved in 43% (24/56) of the entire population and 45% (13/29) of those with TP-53 aberrant disease. Of the 19 patients who achieved uMRD CR and elected to discontinue therapy after cycle 15, only one progression event was observed with a median of 23 months off therapy [45].

The BOVen study was a single-arm phase 2 study that examined 39 frontline CLL patients. The design involved the BTKi zanubrutinib starting in cycle 1, obinutuzumab in cycles 1 to 8, and venetoclax starting in cycle 2. The combination of zanubrutinib and venetoclax was then continued through at least cycle 8 and potentially through cycle 24. The duration of the therapy was determined by the MRD status, which was examined in PB every other cycle starting at cycle 3. When uMRD (10^−4^ sensitivity) was seen in PB, BM uMRD was assessed. If uMRD was confirmed, therapy was discontinued. PB assessment for uMRD was then performed every 3 months as surveillance for those in treatment-free observation. The primary end point of the study was the proportion of patients achieving uMRD in both PB and BM. With a median follow-up of 25.8 months, 33 (89%) patients achieved uMRD in both PB and BM after a median of ten cycles, exceeding the prespecified threshold of 70% to reach the primary end point. At study publication, patients were monitored off treatment for a median of 15.8 months, and 31 of the 33 patients (94%) maintained uMRD [46]. Of the 35 patients to achieve uMRD at 10^−4^ sensitivity, 33 (94%) also had uMRD at the more stringent threshold of 10^−5^. The median PFS was not reached at the time of publication, with only one patient experiencing progressive disease. The combination was generally well tolerated with grade ≥ 3 including neutropenia (18%), thrombocytopenia (8%), rash (8%), and pneumonia (8%). No patients experienced TLS [46].

A novel aspect of BOVen was the analysis of early MRD kinetics as a predictor of which patients may be more likely to respond to a briefer period of therapy. This post hoc analysis examined the change in MRD from the baseline to day 1 of cycle 5, which was selected as patients had received a full month of venetoclax at the target dose. The reduction of MRD to 1/400th of the baseline (*Δ*MRD400) was identified as the optimal cutoff point for predicting uMRD. Of the 21 patients who achieved *Δ*MRD400, 100% had uMRD in BM after six cycles, and 95% (20/21) required <12 cycles of therapy (median eight cycles) to achieve uMRD. By contrast, 14 patients did not reach *Δ*MRD400, and 50% (7/14) required >12 months of therapy (median 13 cycles). Additionally, of the 20 patients who reached *Δ*MRD400 and discontinued therapy, the 1-year recurrent rate of MRD with 10^−5^ sensitivity was 5%. Of the 11 patients who did not achieve *Δ*MRD400 but were able to discontinue therapy after achieving the MRD end point, the 1-year rate of recurrent MRD at 10^−5^ sensitivity was dramatically higher at 75%. This analysis suggests that *Δ*MRD400 may be an effective surrogate to help stratify those who may respond quickly to treatment versus those at risk of disease recurrence [46].

The AVO and BOVen studies both demonstrated very high rates of uMRD. Given that uMRD is thought to predict PFS, these deep responses are encouraging, although longer follow-up will be needed to assess the durability of these regimens. The use of an MRD-guided duration of therapy may allow for a tailored approach to therapy discontinuation in patients who have earlier, deeper responses, minimizing the risks of treatment-related adverse events as well as decreasing the costs of treatment. Given the single-arm nature of these studies, the relative contribution of obinutuzumab to efficacy and toxicity beyond the BTKi and venetoclax combination remains unclear and will be the topic of future study. A summary of key information from the discussed novel–novel CLL studies is included below in Table 2.

## 5. Limitations of Novel–Novel Regimen Data

The current data on targeted therapies have several limitations. First, these doublet and triplet combinations have not been compared with sequential single novel agent therapies, so we do not fully understand how the use of BTKi and BCL2i together compares with sequential monotherapy. While combination regimens have been largely well tolerated, they do appear to have toxicity in excess of single agents. Thus, it is crucial to understand who benefits the most from combination therapy and who would be best served by less toxic sequential monotherapies. When considering doublet vs. triplet regimens, it is important to note that the relative contribution of drugs in combination regimens, particularly of CD20 monoclonal antibodies, is unclear. Finally, while it is considered a promising end point for novel agent-based studies, uMRD has only been clearly proven as a surrogate for survival in CIT-treated patients. Given the excellent outcomes experienced by many CLL patients, survival end points for patients treated with novel agents will take years to mature. Thus, we will not definitively understand the relationship between uMRD and survival, as well as its potential as a criterion for therapy discontinuation for years to come.

## 6. Summary and Future Directions

With the development of targeted therapies, outcomes have dramatically improved for patients with CLL with indisputable improvement in PFS compared with CIT and some evidence of OS benefit. While ibrutinib forged the path for novel agents in CLL, more targeted treatments have demonstrated excellent efficacy with improved safety. Building on the success of single novel agent regimens and given complementary mechanisms of BTKi and venetoclax, doublet and triplet therapies are being explored. The next generation of studies will be instrumental in elucidating the optimal approach to the use of novel–novel combinations. Specifically, head-to-head comparisons of novel–novel combinations and single novel agent regimens as well as doublet vs. triplet regimens to assess the added benefit of CD20 monoclonal antibody will inform future clinical practice. Ongoing studies include CLL17 (NCT04608318), MAJIC (NCT05057494), and ACE-CL-311 (NCT03836261). CLL17 is a three-arm study comparing ibrutinib vs. venetoclax and obinutuzumab vs. I + V. This study will provide insight into the comparative efficacy of two standard of care single novel agent regimens vs. the novel–novel combination I + V. MAJIC is a randomized comparison of acalabrutinib and venetoclax vs. venetoclax and obinutuzumab. This study will similarly compare a novel–novel doublet with a single novel agent standard of care. ACE-CL-311 is a three-arm study comparing AVO, acalabrutinib, and venetoclax and an investigator’s choice of CIT (FCR or BR). While the primary end point of this study is PFS between the AV arm and the CIT arm and the study is not powered to elucidate the contribution of obinutuzumab to the combination, the results from this study should provide insight into the added utility of CD20 monoclonal antibody. With this next series of data, we will better understand how to optimally use combination therapy to maximize efficacy in those most likely to benefit.

## Figures and Tables

**Table 1 cancers-15-01018-t001:** A summary of key information from landmark CLL trials since 2015.

Name	Year Published	Patient Population	Key Findings
RESONATE-2	2015	1L CLL, age > 65	PFS: Ibrutinib > chlorambucil OS: Ibrutinib > chlorambucil
AO41202	2018	1L CLL, age > 65	PFS: Ibrutinib and ibrutinib/rituximab arms > BR OS: No difference bt arms
E1912	2019	1L CLL, age < 70	PFS: Ibrutinib/rituximab > FCR OS: Ibrutinib/rituximab > FCR
UK FLAIR	2021	1L CLL, age < 75	PFS: Ibrutinib/rituximab > FCR OS: No difference bt arms Misc: Fixed duration (12 months)
ILLUMINATE	2019	1L CLL, ≥65 y or <65 y with comorbidites	PFS: Ibrutinib/obinutuzumab > chlorambucil/obinutuzumab OS: No difference bt arms Misc: PFS benefit consistent across high risk features
ELEVATE-TN	2019	1L CLL, ≥65 y or <65 y with comorbidites	PFS: Acalabrutinib/obinutuzumab > CO Acalabrutinib > CO OS: No difference bt arms
ALPINE	2021	R/R CLL	ORR: Zanubrutinib > ibrutinib PFS: Zanubrutinib > ibrutinib
CLL14	2019	1L CLL, CIRS Score > 6	PFS: VO > CO OS: No difference bt arms Misc: Fixed duration (12 months) tx

**Table 2 cancers-15-01018-t002:** A summary of key information from novel–novel trials in CLL.

Drug Combo	Phase	Comparator	Size	NCT #	Primary Endpoint	Results	Notes
I + V (Jain et al. [31])	2	Single arm	80	NCT02756897	Best response (CR/CRi) up to 2 months post I + V	CR/CRi at 1 year in 29/33 (88%); uMRD at 1 year in 45/80 (56%) in BM	Untreated patients ≥ 65 or with HR features
I + V (CAPTIVATE)	2	MRD Cohort: Placebo FD Cohort: Single arm	323	NCT02910583	MRD Cohort: 1 Year DFS in post I + V patients in confirmed uMRD, randomized to I or placebo (n = 86) FD Cohort: CRR (n = 159)	MRD Cohort: 100% I vs. 95% placebo FD Cohort: CRR rate 56%	1L CLL only, age 18–70, FD excluded del(17p)
I + V (GLOW)	3	CO	211	NCT03462719	PFS	PFS at 30 months: 80.5% I + V arm vs. 35.8% CO (*p* < 0.0001)	1L CLL only, ages ≥ 65 or 18-64 with CIRS > 6/CrCl < 70
I + V add-on (Thompson et al [43])	2	Single arm	45	N/A	Rate of uMRD at 1 year	21/33 (64%)	On I for at least 1 year (either as front-line therapy or R/R disease) prior to addition of V, HR features, detectable disease
AVO	2	Single arm	37	NCT03580928	CR with uMRD in BM at 15 months	14/37 (38%)	1L CLL only, age ≥ 18
BOVen	2	Single arm	39	NCT03824483	Rate of uMRD in PB/BM at 1 year	33/37 (89%)	1L CLL only, age ≥ 18

## Data Availability

No new data were created or analyzed in this study. Data sharing is not applicable to this article.

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
