# Peer review of "A History of Targeted Therapy Development and Progress in Novel–Novel Combinations for Chronic Lymphocytic Leukemia (CLL)"

_cancers, 2023, doi:10.3390/cancers15041018_

Round 1

Reviewer 1 Report

This manuscript is about targeted therapies in CLL. The manuscript needs major revision.
1. The manuscript lacks any figures and tables and it is suggested to add them. Consider one of the figures to describe the action mechanism of target therapy in CLL.
2. Check the manuscript completely for English grammar.
3. Many related articles have been published on this topic, which should be referenced in this manuscript. For example:
DOI: 10.1097/PPO.0000000000000416
DOI: 10.1200/EDBK_279099
DOI: 10.1186/s13046-020-01738-0
4. Please consider a section for the prognostic risk score for patients with relapsed or refractory CLL treated with targeted therapies.
5. There is resistance to targeted therapies in CLL. Please consider a section for this issue.
6. Infections in patients with CLL in targeted therapies should be debated.
7. Consider the section on the limitations of targeted therapy in CLL patients.
8. Overcoming methods for resistance to targeted therapies in CLL should be added.
9. Consider the section on the advantage of target therapy should be mentioned.
10. Sectioning of the manuscript is inappropriate. A section on the nature of target therapy is not considered.
11. The introduction is very short.

Reviewer 2 Report

It was a manuscript about the application of targeted therapy for chronic lymphocytic leukemia treatment. Here are some comments on this study that should be considered before publication:

1.      Please improve the quality of the abstract.

2.      please don’t use non-identical abbreviations as keywords.

3.      The introduction section is poor. Please improve it. The same for section 2.

4.      What does “BTK” refer to? Please introduce all the abbreviations at their first-time usage.

5.      Lines 142-157 need to be rewritten.

6.      There are some grammatical mistakes in the text that should be corrected.

7.      Please add a subheading for the limitations and future perspectives.

8.      Please rewrite the conclusion.

Reviewer 3 Report

In this article the authors extensively reviewed targeted therapies in CLL referring to many important clinical trials in recent years. Although the review is well and reasonably written, some points should be improved.

Major comments:

1.       The title of 5. Conclusions should be changed such as “Forthcoming clinical trials” and newly set up “Conclusions” because the authors did not draw definite conclusions in Section 5.  

2.       The authors should briefly mention about adverse events/toxicities in doublet and triplet regimens, especially in triplets, but not only good things.

3.       The 2 sentences in page 8, line 373-376 should be deleted or improved, because the purpose and meanings of these 2 are vague.

Minor comments:

1.       Page 2, line 5: a first→the first

2.       Page 4, line 156: AE should be written in full-term, then abbreviated.

3.       Page 6, line 261: for for→for

4.       Page 6, line 302: Based off→Based on?

Round 2

Reviewer 1 Report

The manuscript is corrected. It is acceptable.

Reviewer 2 Report

Thanks for addressing the comments. 

Reviewer 3 Report

The manuscript has been reasonably revised.